# Immune Response and Metastasis—Links between the Metastasis Driver MACC1 and Cancer Immune Escape Strategies

**DOI:** 10.3390/cancers16071330

**Published:** 2024-03-28

**Authors:** Sebastian Torke, Wolfgang Walther, Ulrike Stein

**Affiliations:** Experimental and Clinical Research Center, Charité, Medical Centre Berlin and Max-Delbrück-Center for Molecular Medicine, 13125 Berlin, Germany; wowalt@mdc-berlin.de (W.W.); ustein@mdc-berlin.de (U.S.)

**Keywords:** metastasis, MACC1, immune evasion

## Abstract

**Simple Summary:**

While both the fields of cancer metastasis research and cancer immunology have been heavily investigated, their combination—the immunology of metastasis—is underrepresented given the fact that metastasis is responsible for up to 90% of cancer deaths. Additionally, evading detection by the immune system is a key prerequisite for the spread of tumor cells to distant organs. In this review, we explore the connections between a master regulator of metastasis, MACC1, and both its direct and indirect links with immunological processes. Specifically, we highlight MACC1-induced alterations of signaling pathways and secreted factors and how this translates into changed immunological outcomes including effects on immune cell infiltration, activity, and their regulation through immunological checkpoints.

**Abstract:**

Metastasis remains the most critical factor limiting patient survival and the most challenging part of cancer-targeted therapy. Identifying the causal drivers of metastasis and characterizing their properties in various key aspects of cancer biology is essential for the development of novel metastasis-targeting approaches. Metastasis-associated in colon cancer 1 (MACC1) is a prognostic and predictive biomarker that is now recognized in more than 20 cancer entities. Although MACC1 can already be linked with many hallmarks of cancer, one key process—the facilitation of immune evasion—remains poorly understood. In this review, we explore the direct and indirect links between MACC1 and the mechanisms of immune escape. Therein, we highlight the signaling pathways and secreted factors influenced by MACC1 as well as their effects on the infiltration and anti-tumor function of immune cells.

## 1. Clinical Significance of MACC1 for Cancer Metastasis

The formation of metastasis remains the most lethal attribute of cancers, being responsible for the majority of cancer-related deaths, and—in some entities—making up the cause of about 90% of cancer deaths [1,2,3]. Although therapeutic strategies are successful in limiting cancer growth and progression, cancer cells will continue to evolve and later form metastases. This highlights the clinical need for the development and evaluation of biomarkers capable of predicting metastasis formation as well as therapy success. In the last ten years, metastasis-associated in colon cancer 1 (MACC1) has been recognized as such a biomarker in over 20 solid cancer entities [4]. It is a causal driver of metastasis, and its level in initially metastasis-free patients is a highly accurate predictor of metastasis formation and overall as well as metastasis-free patient survival. Importantly, multiple studies have so far confirmed that both intra-tumoral MACC1 expression as well as blood levels of MACC1 can serve as a marker predicting metastasis formation [5,6,7,8]. Despite its promising capabilities as a biomarker, MACC1 is so far not widely used in the clinic. Although the role of MACC1 for the initiation and propagation of metastasis has been well-studied, its physiological role in normal cell function remains largely unknown. One study, however, reported that MACC1 defects during embryonal development can lead to malformations [9]. Additionally, MACC1 has been linked to metabolic pathways that mainly regulate glucose and glutamine metabolism [10]. Functionally, MACC1 acts as a transcription factor within the hepatocyte growth factor (HGF)/c-MET signaling pathway, therein relaying extracellular signals [4]. Furthermore, MACC1 is involved in the regulation of cellular functions through many other pathways including mitogen-activated protein kinase kinase (MEK)/extracellular signal-regulated kinases (ERK), phosphoinositide 3-kinases (PI3K)/protein kinase B (AKT)/β-catenin, signal transducers and activators of transcription (STAT)1/3, and twist-related protein (TWIST)/vascular endothelial growth factor (VEGF). Therefore, functionally, MACC1 is involved in mediating cellular proliferation, metabolic activity, cancer cell stemness properties, and angiogenesis, which all lead to the promotion of a metastasis-associated phenotype [4]. Conclusively, it is not surprising that MACC1 is therefore being evaluated as a target of metastasis-specific therapeutic intervention.

While various molecules have been identified that are capable of reducing the expression of MACC1, the most promising approach to date is to use agents from the class of statins. These drugs were initially developed to lower plasma cholesterol levels but have recently shown great potential in inhibiting a MACC1-induced metastatic phenotype in vitro and metastasis in animal models. Moreover, most strikingly, an evaluation of the clinical data from over 300,000 patients including more than 50,000 cancer patients revealed strong evidence of a cancer-preventive effect by statins [11,12]. While all of these highlight the importance and potential that MACC1 possesses as a biomarker for metastasis, additional research is needed to fully elucidate MACC1’s specific physiological functions as well as further understand its role in mediating key cancer processes.

One such process that to date remains only vaguely comprehended is in what manner MACC1 is involved in regulating how tumor cells interact with their environment, and specifically with close-by immune cells. This is of special importance as one key prerequisite for the formation of metastasis is the development of distinct attributes that allow cancerous cells to evade their detection and attack by surveilling immune cells [13]. These immune-evasive features will need to already develop early within the primary tumor, but they become increasingly relevant within the circulation where circulating tumor cells (CTCs) come in close contact with immune cells, and furthermore in the secondary organs in which metastatic cells aim to take root. In this review, we provide a comprehensive overview over the known links between the metastasis-driver MACC1 and the mechanisms of immune evasion.

## 2. MACC1 Correlates with Immune Cell Infiltration

To date, the experimental evidence of MACC1 expression possessing immunological consequences has largely been centered around correlations of MACC1 levels with intra-tumoral immune cell infiltration. The first study, which used bioinformatic analysis of colon adenocarcinoma (COAD) patients based on The Cancer Genome Atlas (TCGA), identified MACC1 as a positive regulator of the infiltration of natural killer (NK) cells, macrophages, and neutrophils. However, this study could not ascertain any effects on the levels of infiltrating dendritic cells, B cells and T helper, cytotoxic, or regulatory T cells [14]. A second study in breast cancer confirmed the positive effect of MACC1 on the infiltration of macrophages, in this case, specifically of anti-inflammatory, tumor-associated macrophages (TAMs). Here, however, the effect on NK cell infiltration was negatively associated with MACC1, as was the infiltration of cluster of differentiation (CD)8+ cytotoxic T lymphocytes (CTLs) [15]. Of note, one study reported that the levels of MACC1 were closely related with the expression of the immunological checkpoint programmed cell death 1 ligand 1 (PD-L1). Here, the ectopic manipulation (up- or downregulation) of MACC1 translated into corresponding changes of PD-L1, and ultimately into alterations of the anti-tumor effects and immune cell-mediated killing capacity of peripheral blood mononuclear cells (PBMCs) in a co-culture setting [16]. Although this provides the first experimental evidence of the correlation between MACC1 and the immune system, further links can be drawn between MACC1 target genes and immune cell infiltration and function.

## 3. MACC1 Influences Immune Cell Infiltration and Tumor-Immunity through Positive Feedback Loop and Vascularization

One of the main functions of MACC1 is to act as a transcription factor. In this regard, a number of its target genes have been described to possess the potential for inducing effects of immunological relevance. First and foremost, in a positive feedback loop following the activation of MACC1 downstream of HGF/c-Met, MACC1 induces the expression of c-Met [17,18]. c-Met itself possesses various properties related to the immune system. Most notably, HGF stimulation can induce the expression of PD-L1 through c-MET and thereby contribute to cancer cell immune evasion [19,20], a process likely dependent on MACC1 as a core regulator/effector of this signaling pathway. Moreover, HGF/c-Met regulates the recruitment of immune cells into the tumor microenvironment (TME) and can create an immunosuppressive milieu (e.g., by inducing anti-inflammatory T helper cells and macrophages as well as shifting the cytokine milieu toward the secretion of immune system-regulating cytokines). Specifically, HGF-stimulation can diminish the production of IFN-γ, whilst interleukin (IL)-4 and IL-10 secretion increases [19,21,22].

Additionally, one of the functional outcomes of HGF/c-Met stimulation is the MACC1-mediated induction of angiogenesis through increasing the production and secretion of VEGF [23,24]. VEGF, in turn, can directly influence the recruitment, differentiation, and activity of cells belonging to the innate as well as adaptive immune system. Specifically, the secretion of VEGF leads to the specific recruitment of a variety of regulatory immune cells such as regulatory T cells (Treg) and myeloid-derived suppressor cells (MDSCs). Furthermore, VEGF pushes macrophage differentiation into an anti-inflammatory M2-subset. Additionally, VEGF influences the maturation of dendritic cells (DCs) and increases their expression of immuno-regulatory molecules such as PD-L1, which act as immune checkpoints and can limit the activation and function of other immune cells in close proximity to those DCs. All of these mechanisms promote an anti-inflammatory milieu and a pro-tumor environment. Additionally, VEGF can directly influence the proliferation, recruitment, and cytotoxicity of killer cells such as CTLs and NK cells, decreasing their recognition and effector functions against cancer cells [25,26]. Notably, MACC1 increases the vessel density and vascularization through VEGF/Twist. However, VEGF also influences the expression of adhesion molecules on endothelial cells, specifically blocking the upregulation of molecules such as intercellular adhesion molecules (ICAMs), vascular cell adhesion protein (VCAM), or selectins that are induced by inflammatory mediators. This decreases the extravasation and tumor infiltration of various immune cells normally mediated by those adhesion molecules. Additionally, VEGF can also specifically increase other endothelial adhesion molecules such as CLEVER-1 and Fas-ligand (FasL). CLEVER-1 is a scavenging receptor and correlates with the selectively increased infiltration of anti-inflammatory and pro-tumor immune cells such as Tregs and M2 macrophages. Endothelial expression of FasL initiates apoptosis pathways specifically in activated T cells, with Tregs being resistant to FasL-mediated killing. Ultimately, this is associated with decreased numbers of intra-tumoral CTLs [27]. Interestingly, a direct link has already been described between the expression of MACC1 and the Fas/FasL apoptosis pathway.

## 4. MACC1 Mediates Immune Evasion through STAT1/3 and Fas

The tumor-intrinsic expression of MACC1 is relevant for the sensitization to death receptor-mediated apoptosis via Fas/FasL. In this context, the presence of MACC1 increases the activation and phosphorylation of STAT proteins that induce the expression of myeloid cell leukemia-1 (MCL1), which in turn decreases the expression of Fas [4]. This effect can protect cancer cells from Fas-agonist induced apoptosis induction and likely, in a more physiological context, from immune cell-induced destruction. Interestingly, STAT proteins can also mediate other mechanisms of immune evasion. Of note, the STAT signaling pathway—when induced by interferon (IFN)-γ—can drive the overexpression of PD-L1, thereby inhibiting the anti-cancer functions of tumor-targeting immune cells [28]. Interestingly, there is already a reported connection between MACC1 and PD-L1. In gastric cancer tissues, the levels of MACC1 were positively correlated with the expression of PD-L1. Furthermore, alterations in the expression of MACC1 through genetic manipulation such as silencing or induced overexpression translated into corresponding changes of PD-L1 expression. However, it is unclear which cells within the TME are expressing PD-L1 in this study. Therefore, additional investigations are needed to elucidate whether there is a direct link of MACC1 and the expression of immunological checkpoints [16]. In a more general context, STAT1 expression not only correlates with the expression of PD-L1 and its interaction partner programmed cell death protein 1 (PD-1), but can also be associated with disease stage and tumor grade [29]. Moreover, one transcriptional target of STAT proteins is c-Myc, which can be induced by STAT3 in early tumorigenesis and in response to T cell immunosurveillance. The induction of c-Myc can then drive immunoediting, leading to decreased T cell recognition and generally an immune-suppressive TME [30]. However, STAT proteins are also responsible for the regulation of the expression of major histocompatibility complex (MHC) molecules induced by IFN-γ. Therein, they are centrally involved in how cancer cells present antigens and how they are recognized and targeted by immune cells. Importantly, many cancers dysregulate pathways of antigen presentation, however, the used strategies for achieving this are plentiful. Overall, the outcome of STAT protein expression and function can vary, even within a specific cellular subset. However, if pro- or anti-tumor functions predominate, depends on the individual context as well as additional contributing factors [31,32]. Finally, as STAT proteins are major immune regulators, one key aspect is that they can induce the production of cytokines by cancer cells [33,34].

## 5. The Role of Cytokines and Stemness Factors for MACC1 and the Immune System

In this context, cancer cells can use inflammatory mediators such as cytokines to promote their own growth. Moreover, chronic inflammatory conditions can initiate the aberrant changes leading to cancer formation, and these conditions can also continuously drive cancer progression. MACC1, in particular, has been identified as one target initiated through the tumor-promoting effects of two major regulators: the cytokines tumor necrosis factor (TNF)-α and IFN-γ [35]. Here, the signaling cascades activated by the cytokines lead to the activation of nuclear factor ‘kappa-light-chain-enhancer’ of activated B-cells (NFkB), resulting in the induction of transcription factors binding to the MACC1 promoter and thereby to a shift toward a metastasis-associated phenotype mediated by MACC1 itself. Although these cytokines naturally influence many cells within the TME, this is an interesting example of how the inflammatory properties surrounding cancerous cells can contribute to metastasis. Interestingly, large amounts of various cytokines can be produced by cancer stem cells [36], which have already been linked to MACC1.

While cancer stem cells (CSCs) only make up a small fraction of the total tumor cells, they likely provide the basis for the majority of malignant human tumors and are critically important in initiating metastasis [37,38]. In this context, MACC1 has been identified to regulate the cancer cell stemness properties through the induction of transcription factors known as core regulators of multipotency such as octamer-binding transcription factor 4 (Oct4) and Nanog [39]. Importantly, CSCs possess a variety of strategies to avoid immune recognition. Most notably, they employ immune checkpoint molecules to evade cell contact-induced killing by immune cells and can secrete extracellular vesicles (EVs), growth factors, metabolites, and cytokines to modulate the TME in a wider scope [40]. Specifically, CSCs exhibit high levels of expression for the inhibitory molecules PD-L1, cytotoxic T-lymphocyte-associated protein 4 (CTLA-4), T-cell immunoglobulin, and mucin-domain containing-3 (TIM-3) as well as lymphocyte-activation gene 3 (LAG3). Additionally, CSCs often downregulate the expression of MHC class I molecules in comparison to more differentiated cancer cells, a feature that conveys their protection from T cell recognition and destruction. Interestingly, CSCs are also capable of reducing the expression of tumor-associated antigens, thereby additionally limiting immune cell surveillance [41]. Furthermore, CSCs increase the expression of CD47, a transmembrane protein that transmits inhibitory signals toward macrophages and hinders phagocytosis, thereby acting as an immune checkpoint for the innate immune system [42]. Through the secretion of soluble factors, CSCs can further manipulate their surroundings. EVs, for instance, are capable of suppressing T cell functions, inhibiting dendritic cell maturation, and pushing macrophages into the anti-inflammatory M2 subtype [41]. In addition, CSCs secrete a variety of cytokines including IL-4, IL-8, transforming growth factor (TGF)-β, macrophage colony-stimulating factor (M-CSF), and granulocyte-macrophage colony-stimulating factor (GM-CSF), all possessing immunosuppressive capabilities and collaborating to produce a pro-tumor environment through the manipulation of the anti-tumor activity of immune cells [42]. Importantly, these functional outcomes can be directly linked to core multipotency factors such as Oct4 [43], which in turn can be induced by active MACC1 [39]. Along the same lines, Nanog conveys this immune-resistant phenotype and can also be upregulated by MACC1 [39,44]. Finally, MACC1 can induce stemness properties by stimulating LGR5 expression [45].

## 6. MACC1 Manipulates the TME via PI3K/Akt and Wnt Pathways

Another link of MACC1 with mechanisms of immune evasion is mediated through the regulation of the PI3K/AKT pathway. Various studies have identified MACC1 as a positive regulator of PI3K/AKT signaling [46,47], at least in part through a negative regulation of the phosphatase and tensin homolog (PTEN) [4,48,49]. Activation of this pathway can create an inhibitory myeloid TME and reduce the overall numbers of infiltrating CTLs [50]. Furthermore, through the loss of activity from PTEN, specific changes in the cytokine milieu such as a high production of chemokine (C-X-C motif) ligand (CXCL)17 can induce the recruitment of MDSCs, Tregs, and anti-inflammatory M2 macrophages. Moreover, the loss of PTEN or its secreted variant PTEN-L can diminish the production and release of pro-inflammatory cytokines such as TNF-α and IL-6. Importantly, reduced PTEN-activity is also linked to an increased expression of immunological checkpoints such as PD-1 and PD-L1 [51]. All of these described alterations collaborate to create an immunosuppressive environment as well as overall favorable conditions for continuous tumor cell growth.

One of the outcomes of increased AKT kinase activity is the stabilization of β-catenin and the promotion of its nuclear translocation, thereby connecting the PI3K/Akt pathway closely with the Wnt/β-catenin pathway. Various studies have already linked MACC1 with the Wnt pathway [52]. In this context, MACC1 has been established as a transcriptional target of Wnt/β-catenin [53], and more importantly, MACC1 can regulate the expression of Wnt target genes such as VEGF, c-Myc, cyclin D1/E, and matrix metalloproteinases [4,39]. The activation of Wnt signaling in tumor cells can then, in turn, promote immune evasion through the direct induction of PD-L1 and CD47 expression, and furthermore via an alteration of the infiltration and function of immune cells within the TME. Specifically, increased levels of Wnt-induced secreted protein 1 (WISP1) facilitate a pro-tumor microenvironment by promoting TAMs whereas the downregulation of chemokines such as chemokine (C-C motif) ligand (CCL)4 and CCL5 reduce the numbers of DCs [54,55]. Additionally, an inverse correlation has been described between Wnt/β-catenin signaling activity and the infiltration of T cells into cancerous tissue [56]. One key target gene of this signaling pathway is S100A4, also referred to as *metastasin*. This master regulator of metastatic functions including epithelial–mesenchymal transition (EMT) [57] has been shown to be a transcriptional target of MACC1 [5]. S100A4 can stimulate tumor cells to secrete numerous inflammatory cytokines, most notably IL-8 and CCL2, which then shape the TME toward a pro-tumor state and favor metastatic growth [58]. Furthermore, S100A4 is also positively linked with the expression of PD-L1 and the inhibition of anti-tumor T cell activity. Of note, this is not only a direct effect, but can also act on the surrounding cells, as S100A4 can be transmitted (e.g., via extracellular vesicles) [59]. Interestingly, MACC1 and S100A4 have been independently established as key drivers of metastasis and can individually be used to stratify patients according to their specific metastatic risk. However, combining both markers yields the greatest potential for patient stratification. Additionally, therapeutic strategies have been developed to target both MACC1 as well as S100A4. Here, again, combining the interventional therapies works synergistically and has the highest therapeutic value in vitro and in limiting metastasis formation in animal models [5].

## 7. Additional Mechanisms by Which MACC1 Might Influence Immune Cell Infiltration and Function

Interestingly, MACC1 can also influence metastasis and immune evasion via a rather unexpected pathway—by modulating the mechanisms of endocytosis. MACC1 has been described to promote receptor internalization and recycling as part of clathrin-mediated endocytosis (CME) [60]. In the context of metastasis, the stimulation or continuation of CME can facilitate stronger or longer lasting signals from growth factor receptors such as the EGFR, thereby stabilizing oncogenic signaling [61]. Interestingly, endocytic pathways can also influence tumor immunity and tumor-induced immune suppression through the regulation of tumor surface antigens. Here, especially the degradation of molecules involved in antigen presentation, namely MHC class I, is of considerable importance [62,63]. Endocytosis, as a pathway that is based around membrane-vesicles deemed for the degradation and recycling of intra- and extracellular components, is furthermore mechanistically closely related to another pathway, autophagy [64]. Importantly, MACC1 has been described as a positive regulator of 5′ adenosine monophosphate-activated protein kinase (AMPK)/Unc-51-like autophagy-activating kinases (ULK1) induced autophagy. Tumor cells employing this tool can reroute MHC class I molecules toward lysosomes, leading to their destruction and an overall diminished T cell recognition. Through autophagy, tumor cells can also alter the secretion of cytokines into the TME, affecting, for instance, CXCL1, CXCL2, CXCL5, and CXCL12 production. This ultimately leads to an attraction of MDSCs and immunosuppressive macrophages as well as an overall pro-tumor environment. Furthermore, autophagy diminishes the release of the chemokine CCL5, which facilitates the infiltration of NK cells into the TME. Additionally, through autophagic processes, tumor cells become capable of degrading granzyme B, a factor secreted by activated CTLs and NK cells with the aim to induce apoptosis in target cells, thereby effectively blocking immune cell induced tumor cell destruction [65]. Autophagy is also a process providing additional energy when the metabolism of a cell is challenged [66]. Interestingly, there is a more direct link described between MACC1 and the metabolic system.

First and foremost, MACC1 is capable of altering important metabolic pathways, leading to elevations in glucose and glutamine uptake. In this way, MACC1 functions through the upregulation of specific transporter proteins such as glucose transporter (GLUT)1 and GLUT4 as well as via the regulation of glycolytic enzymes including hexokinase, pyruvate dehydrogenase kinase, and lactate dehydrogenase [10,67]. Through the induction of these metabolic changes—namely by promoting the so-called Warburg-effect, leading to an increased use of glucose—cancer cells promote nutrient depletion, oxygen-deprivation, increased acidity, and the release of potentially toxic metabolites into their surrounding area [68]. These changes promote a TME that is highly immunosuppressive, hinders immune cell infiltration (immune exclusion), and can even mediate the loss of efficacy of adoptive cell therapies and immune checkpoint inhibitors [69]. Additionally, it has been described that hypoxia can induce the expression of PD-L1 on various cell types of the TME including tumor cells. Furthermore, the combination of glucose and oxygen deprivation can diminish the overall antigen presentation on cancer cells via the MHC class I system [70,71].

Although also linked to metabolism but independent from cancer formation, a relationship between MACC1 and obesity has been described. In this study, higher levels of cell-free MACC1 in plasma were observed in obese adults in comparison to normal weight control individuals. Reducing the total body fat resulted in lowered MACC1 levels. Additionally, rats with high-fat diet-induced obesity demonstrated higher MACC1 levels and a more severe colon tumor outcome [72]. Importantly, obesity can promote chronic inflammation and induce systemic changes in T cell and macrophage populations, exacerbating various diseases including cancer [73].

## 8. Outlook and Conclusions

Since its first description in 2009, MACC1 has been recognized as a prognostic and predictive biomarker as well as a driver of metastasis formation in over 20 cancer entities. MACC1 levels analyzed from tumor tissue or liquid biopsies can be used to determine the individual metastatic risk and predict patient survival and therapy outcomes. However, there is only limited data of MACC1 as a biomarker of immune system status and anti-tumor immunity. Several studies have identified correlations between MACC1 expression levels and the infiltration of immune cells into the tumor surrounding areas [14,15]. Furthermore, a bioinformatic analysis of COAD using multiple databases identified MACC1 as a potential predictor for immune responses as well as a novel target for immunotherapy-based intervention strategies [74]. Further evidence is provided by the general use of statins, potent transcriptional inhibitors of MACC1 expression, as they are widely used and are often also prescribed to patients treated for cancer. Here, multiple studies have reported that statins are not only capable of preventing cancer development [12], but also augment already established immune-targeting treatment regimens. In this case, statins can enhance therapy responses when targeting immunological checkpoints such as PD-L1, a factor that can be regulated in its expression by MACC1 [16,75,76,77]. However, direct experimental links are still rare and future studies need to provide additional and more extensive connections between MACC1 as a metastatic switch and its impact on the immune system. Here, the description of a direct link between intra-tumoral MACC1 expression and the state/function of nearby immune cells would especially help further the understanding of the immunological consequences of MACC1. Additionally, it would be interesting to study whether there is any difference in the MACC1 expression levels between immune checkpoint therapy responders or non-responders.

In this review, we have demonstrated that the intra-tumoral expression of MACC1 facilitates cancer immune evasion through multiple processes (Figure 1, Table 1), all leading toward the altered expression of immune system-controlling surface molecules or cytokines, ultimately inducing changes in the composition and function of the TME. While MACC1 is already established as a marker of metastasis, we highlight here that it possesses additional value as a predictor for cancer immune evasion by manipulating immune cell infiltration and function. Furthermore, the points made in this review indicate that there is a close relationship between metastasis and immune evasion that has been largely underestimated in the past. Here, additional work should be performed to analyze the links between molecules associated with or causally driving metastasis and their potential immunological consequences. Taken together, we have demonstrated that MACC1 shapes the environment for tumor cells to promote the formation of metastasis by mediating various strategies relevant to immune escape mechanisms of cancer cells.

## Figures and Tables

**Figure 1 cancers-16-01330-f001:**
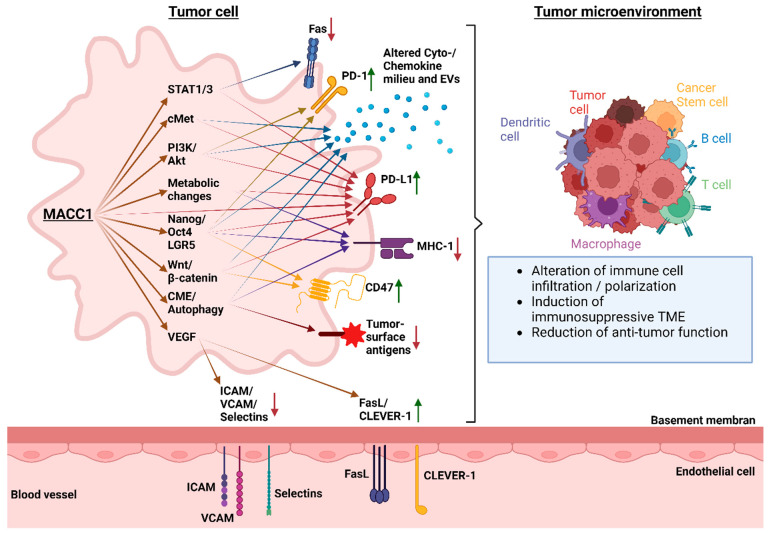
Links between MACC1 and mechanisms of immune escape. MACC1 directly affects the expression or function of the indicated factors, which in turn leads to immunological consequences marked by the arrows. Up- (green arrow pointing upward) or downregulation (red arrow pointing downward) are next to the immunologically relevant markers. Consequently, this affects the tumor microenvironment and creates a tumor-favorable milieu.

**Table 1 cancers-16-01330-t001:** Immunological consequences of MACC1 expression. Upwards-pointing arrows indicate an upregulation or increase, downwards-pointing arrows a downregulation or decrease.

MACC1 Effect	Direct/Indirect Consequences	Effect on Immune System	Investigated Entity	References
**MACC1 expression itself**	PD-L1 ↑	Reduced anti-tumor immune function	Gastric cancer	[16]
**Positive feedback to HGF/c-Met**	PD-L1 ↑ Anti-inflammatory TME (Th cells, macrophages) Immune system- regulating cytokines ↑	Altered immune cell infiltration Reduced anti-tumor immune function	Renal cancer Autoimmune diseases	[19,20,21,22]
**Induction of VEGF**	ICAM, VCAM, Selectins ↓ CLEVER-1, FasL ↑	Specific recruitment of Tregs and MDSCs M2 macrophage polarization Recruitment/anti-tumor effect of CTLs/NK cells ↓	Endothelial cells Breast cancer	[25,26,27]
**Activation of STAT1/3**	Fas expression ↓ PD-L1 ↑ C-Myc ↑	Protection against receptor-induced apoptosis Reduced anti-tumor immune function Immunoediting ↑ Immuno-suppressive TME	Renal cancer Oral/gastric cancer Skin cancer Lung cancer Breast cancer	[28,30,31,32,33,34]
**Facilitation of cancer stemness via Oct4 and Nanog**	Immune checkpoints ↑ MHC class I ↓ CD47 ↑ Cyto-/Chemokines and EVs to modulate TME	Immuno-suppressive TME Reduced anti-tumor immune function Evasion of T cell recognition	Colorectal cancer Brain cancer Breast cancer Colon cancer Head and neck cancer	[36,39,40,41,42,43,44]
**PI3K/Akt signaling**	PTEN ↓ Changed cyto-/chemokine milieu PD-1/PD-L1 ↑	Inhibitory myeloid TME Infiltration of CTLs ↓ Recruitment of MDSCs, Tregs and TAM	Skin cancer Lung cancer Breast cancer Colorectal cancer Brain cancer	[50,51]
**Wnt signaling**	Wnt target genes (VEGF, c-Myc, cyclin D1/E, MMPs) PD-L1/CD47 ↑ Changed cyto-/chemokine milieu Induction of S100A4	Reduced infiltration of DCs and CTLs Promoting TAMs	Brain cancer Skin cancer Lung cancer	[54,55,56,57,58,59]
**Endocytosis and autophagy**	Stronger/longer signals from growth factor receptors Tumor surface antigen ↓ MHC class I ↓ Changed cyto-/chemokine milieu	Evasion of T cell recognition Pro-tumor environment characterized by MDSCs and TAMs Infiltration of NK cells ↓ Degradation of granzyme B → protection from immune cell destruction	Pancreatic cancer Lung cancer	[62,64,65]
**Metabolism**	Glucose transporters/glycolytic enzymes ↑ PD-L1 ↑ MHC class I ↓	Nutrient depletion, oxygen-deprivation, increased acidity, and accumulation of metabolites Highly immunosuppressive TME Immune exclusion	Breast cancer Lung cancer Fibrosarcoma Skin cancer	[68,69,70,71]

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
