# Peer review of "Immune Response and Metastasis—Links between the Metastasis Driver MACC1 and Cancer Immune Escape Strategies"

_cancers, 2024, doi:10.3390/cancers16071330_

Round 1

Reviewer 1 Report

Comments and Suggestions for Authors

The present review describes the effect of MACC1 in the regulation and modulation of the immune response within the tumor microenvironment. The subject is well described and discussed, with an overall look on the main processes and mechanisms of immune escape in cancer.

However, I would suggest some minor revision to improve the quality of the manuscript.

- Any reference to the table and figure is missing in the text

- I would add a column in table 1 describing the type of tumor in which each MACC1 effects and consequences have been studied

- Figure 1 is redundant with the table and is not self-intuitive. I would remove it, as it does not bring any additional information or clarification.

Author Response

Please see the attached response letter.

Reviewer 2 Report

Comments and Suggestions for Authors

1. Title: "The immunology of metastasis" is too broad. Does it regulate metastasis? Please be more definitive.

2. Abstract: Is MACC1 a driver of metastasis? The immune system can evoke an effective antitumor response , but the sole presence of anti-tumor T cells or high NK cell numbers in cancer patients do not guarantee protective immunity and the cancer immunity cycle is frequently hampered in cancer patients. The inhibition of the anti-tumor immune response through the orchestration of an immunosuppressive microenvironment, which is a crucial mechanism that contributes to cancer progression and metastasis. Importantly, tumor-educated
immune cells can promote metastasis by additional means that go beyond the suppression of anti-tumor immunity. Mechanisms? The (pre-)metastatic niche and immune cells as key coordinators of ‘fertile soil’. Inhibiting tumor-promoting inflammation to fight metastatic disease. Tumors of different types, and even individual tumors of the same type, vary greatly in their immune landscape, both locally and systemically.  What are the different factors that regulate the sensitivity of organ-specific metastases versus primary tumors to
immunomodulation?

3. Introduction: Does MACC1 involve the HIPPO signaling pathway? What role do epigenetic programming and MACC1 play in the promotion of metastasis?

4. What role does H. pylori play in MACC1 remodeling of GI cancers?

5. The review should have graphical figures outlining the MACC1 role in metastasis.

Comments on the Quality of English Language

The review reveal over 250 grammar error with 9% plagiarism.

Author Response

(The authors gave the same response as above.)

Round 2

Reviewer 2 Report

Comments and Suggestions for Authors

Revised manuscript is acceptable for publication

Comments on the Quality of English Language

minor editorial edits